# Exploring the interpretability of LSTM neural networks over multi-variable data

## Abstract

In learning a predictive model over multivariate time series consisting of target and exogenous variables, the forecasting performance and interpretability of the model are both essential for deployment and uncovering knowledge behind the data. To this end, we propose the interpretable multi-variable LSTM recurrent neural network (IMV-LSTM) capable of providing accurate forecasting as well as both temporal and variable level importance interpretation. In particular, IMV-LSTM is equipped with hidden state matrix and update process, so as to learn variables-wise hidden states. On top of it, we develop a mixture attention mechanism and associated summarization methods to quantify the temporal and variable importance in data. Extensive experiments using real datasets demonstrate the prediction performance and interpretability of IMV-LSTM in comparison to a variety of baselines. It also exhibits the prospect as an end-to-end framework for both forecasting and knowledge extraction over multi-variate data.

## 1 Introduction

Our daily life is now surrounded by various types of sensors, ranging from smart phones, video cameras, Internet of things, to robots. The observations yield by such devices over time are naturally organized in time series data (Qin et al., 2017; Yang et al., 2015). In this paper, we focus on multi-variable time series consisting of target and exogenous variables. Each variable corresponds to a monitoring over physical world. A predictive model over such multi-variable data aims to predict the future values of the target series using historical values of target and exogenous series.

In addition to forecasting, the interpretability of prediction models is essential for knowledge extraction, variable selection and so on (Hu et al., 2018; Foerster et al., 2017; Lipton, 2016). For multi-variable time series in this paper, we focus on two types of importance interpretation. (1) Variable-wise temporal importance: exogenous variables present different temporal influence on the target one (Kirchgässner et al., 2012). For instance, for the exogenous variable instantaneously correlated to the target one, its historical data at short time lags is expected to high importance values. (2) Overall variable importance: exogenous variables and the auto-regressive part of the target variable differ in predictive power, which reflects different variable importance w.r.t. the prediction of the target (Feng et al., 2018; Riemer et al., 2016). The ability to unveil such knowledge through predictive models enables to fundamentally understand the relevance of exogenous variables to the target one.

Recently, recurrent neural networks (RNNs), especially long short-term memory (LSTM) (Hochreiter & Schmidhuber, 1997) and the gated recurrent unit (GRU) (Cho et al., 2014), have been proven to be powerful sequence modeling tools in a variety of tasks such as language modelling, machine translation, health informatics, time series, and speech (Ke et al., 2018; Lin et al., 2017; Lipton et al., 2015; Sutskever et al., 2014; Bahdanau et al., 2014).

However, current RNNs fall short of the aforementioned interpretability for multi-variable data due to their opaque internal states. Specifically, when fed with the multi-variable observations of the target and exogenous variables, RNNs blindly blend the information of all variables into memory cells and hidden states which are used for prediction. It is intractable to distinguish the contribution of individual variables into the prediction through hidden states (Zhang et al., 2017). Recently, attention-based neural networks have been proposed to enhance the ability of RNN in selectively using long-term memory and the interpretability (Vaswani et al., 2017; Qin et al., 2017; Choi et al.,

2016; Vinyals et al., 2015; Chorowski et al., 2015; Bahdanau et al., 2014). Nevertheless, current attention mechanism is mostly applied to hidden states across time steps and capture globally temporal information, thereby failing to uncover fine-grained variable level importance.

To this end, in this paper we build interpretable LSTM models with the aim to achieve a unified framework of forecasting and knowledge extraction. In particular, the contribution is fourfold. First, we propose the interpretable multi-variable LSTM, referred to as IMV-LSTM, with hidden state matrix and updating scheme, such that each element of the hidden state matrix encodes information for a certain input variable. Second, based on these variable-wise hidden states we develop a novel mixture temporal and variable attention mechanism. Third, attention values are further summarized to quantify variable-wise temporal importance and overall variable importance. Lastly, we perform extensive experimental evaluation of IMV-LSTM against statistical, machine learning and neural network baselines to demonstrate the superior prediction performance and interpretability of IMV-LSTM. The idea of IMV-LSTM easily applies to other RNN structures, e.g. GRU and stacked recurrent layers. This will be the future work.

## 2 RELATED WORK

In time series analysis, prediction with exogenous variables can formulated as an auto-regressive exogenous model or prediction modeles defined on exogenous variables. Vanilla RNNs have been used to study it in (Zemouri et al., 2010; Diaconescu, 2008), where interpretability was not investigated.

Recent research on the interpretability of RNNs is categorized into two groups: attention methods and post-analyzing over trained models. Attention mechanism has gained tremendous popularity (Xu et al., 2018; Choi et al., 2018; Guo et al., 2018; Lai et al., 2017; Qin et al., 2017; Cinar et al., 2017; Choi et al., 2016; Vinyals et al., 2015; Bahdanau et al., 2014). However, current attention mechanism is mainly applied to hidden states across time steps. (Qin et al., 2017; Choi et al., 2016) use weighted input data learned by encoder networks to do forecasting. Weighting input data by attention does not consider the direction of correlation with the target. Moreover, this attention is derived from the hidden states encoding all input variables and thus each element of the attention is composed of contributions from all input variables. Using such attention to interpret variabel importance is biased. It fails short of interpretability on variable-wise temporal importance as well.

As for post-analyzing based interpretation, (Murdoch et al., 2018; Murdoch & Szlam, 2017; Arras et al., 2017) extracted temporal importance scores over words or phrases of individual language sequences by decomposing the memory cells of trained LSTM. (Chu et al., 2018) proposed interpretation solutions for piece-wise linear neural networks. In (Wang et al., 2018), it quantified the importance of each middle layer to the output. (Foerster et al., 2017) introduced input-switched affine transformations into RNNs, which analyzed the contribution of input steps via linear methods. Above work focuses on global temporal importance and does not support variable specific temporal interpretation. Our following proposed IMV-LSTM is a combination of attention and post-analyzing for multi-variable time series, where novel variable-wise hidden states and mixture attention enable fine-grained temporal and variable importance interpretation during the training.

Another line of related research is about tensorization and selectively updating of hidden states in RNNs. (Do et al., 2017; Novikov et al., 2015) proposed to represent hidden states as a matrix. (He et al., 2017) developed tensorized LSTM to enhance the capacity of networks without additional parameters. (Kuchaiev & Ginsburg, 2017; Neil et al., 2016; Koutnik et al., 2014) put forward to partition the hidden layer into separated modules with different updates. The hidden state tensors and update processes in existing works do not maintain variable-wise correspondence, thereby lacking the desirable interpretability.

## 3 INTERPRETABLE MULTI-VARIABLE LSTM

Assume we have $N$-1 exogenous time series and a target series $\mathbf{y}$ of length $T$, where $\mathbf{y} = [y_1, \cdots, y_T]$ and $\mathbf{y} \in \mathbb{R}^T$.[1] By stacking exogenous time series and target series, we define a multi-variable input series as $\mathbf{X}_T = \{\mathbf{x}_1, \cdots, \mathbf{x}_T\}$, where $\mathbf{x}_t = [\mathbf{x}_t^1, \cdots, \mathbf{x}_t^{N-1}, y_t]$. Both of $\mathbf{x}_t^n$ and $y_t$ can be multi-dimensional vector. $\mathbf{x}_t \in \mathbb{R}^N$ is the multi-variable input at time step $t$. Given $\mathbf{X}_T$, we aim to

---

[1]Vectors are assumed to be in column form throughout this paper.

learn a non-linear mapping to predict the next values of the target series, namely $\hat{y}_{T+1} = \mathcal{F}(\mathbf{X}_T)$. Meanwhile, through trained model $\mathcal{F}(\cdot)$, we aim to extract variable-wise temporal importance and overall variable importance w.r.t. the prediction from the data. The following described IMV-LSTM can be easily extended to multi-step ahead prediction via iterative methods as well as vector regression (Fox et al., 2018; Cheng et al., 2006).

## 3.1 NETWORK ARCHITECTURE

In IMV-LSTM we develop hidden state matrix and update scheme, which ensure that each element of the hidden state matrix encapsulates information exclusively from a certain variable of the input. It gives rise to mixture attention on both variable and temporal aspects and fine-grained interpretation described below.

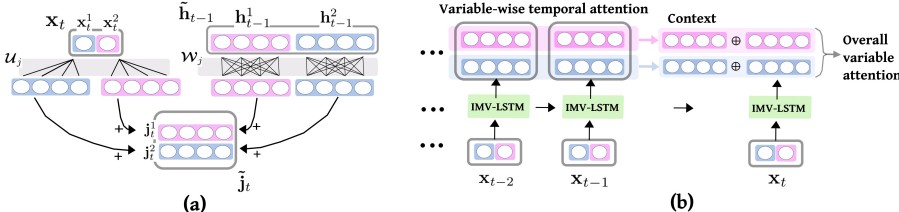

Figure 1: A toy example of a IMV-LSTM with a two-variable input sequence and the hidden matrix of $4$-dimensions per variable. Circles represent one dimensional elements. Purple and blue colors correspond to two variables. Blocks containing rectangles with circles inside represent input data and hidden matrix. Panel (a) exhibits the derivation of hidden update $\tilde{\mathbf{j}}_t$. Grey areas represent transition weights. Panel (b) demonstrates the mixture attention process. (best viewed in color)

To distinguish from the hidden state and gate vectors in a standard LSTM, hidden state and gate matrices in IMV-LSTM are denoted with tildes. Specifically, we define the hidden state matrix at time step $t$ as $\tilde{\mathbf{h}}_t = [\mathbf{h}_t^1, \cdots, \mathbf{h}_t^N]^\top$, where $\tilde{\mathbf{h}}_t \in \mathbb{R}^{N \times d}$, $\mathbf{h}_t^n \in \mathbb{R}^d$. The overall size of the layer is derived as $D = N \cdot d$. The element $\mathbf{h}_t^n$ of $\tilde{\mathbf{h}}_t$ is the hidden state vector specific to $n$-th input variable.

Then, we define the input-to-hidden transition as $\mathcal{U}_j = [\mathbf{U}_j^1, \cdots, \mathbf{U}_j^N]^\top$, where $\mathcal{U}_j \in \mathbb{R}^{N \times d \times d_0}$, $\mathbf{U}_j^n \in \mathbb{R}^{d \times d_0}$ and $d_0$ is the dimension of individual variables at each time step. The hidden-to-hidden transition is defined as: $\mathcal{W}_j = [\mathbf{W}_j^1, \cdots, \mathbf{W}_j^N]$, where $\mathcal{W}_j \in \mathbb{R}^{N \times d \times d}$ and $\mathbf{W}_j^n \in \mathbb{R}^{d \times d}$.

As standard LSTM neural networks (Hochreiter & Schmidhuber, 1997), IMV-LSTM has the input $\mathbf{i}_t$, forget $\mathbf{f}_t$, output gates $\mathbf{o}_t$ and the memory cells $\mathbf{c}_t$ in the update process. Given the newly incoming input $\mathbf{x}_t$ at time $t$ and the hidden state matrix $\tilde{\mathbf{h}}_{t-1}$, the hidden state update is defined as:

$$\tilde{\mathbf{j}}_t = \tanh\left(\mathcal{W}_j \circledast \tilde{\mathbf{h}}_{t-1} + \mathcal{U}_j \circledast \mathbf{x}_t + \mathbf{b}_j\right), \tag{1}$$

where $\tilde{\mathbf{j}}_t = [\mathbf{j}_t^1, \cdots, \mathbf{j}_t^N]^\top$ has the same shape as hidden state matrix (i.e. $\mathbb{R}^{N \times d}$). Each element $\mathbf{j}_t^n \in \mathbb{R}^d$ corresponds to the update of the hidden state w.r.t. input variable $n$. Term $\mathcal{W}_j \circledast \tilde{\mathbf{h}}_{t-1}$ and $\mathcal{U}_j \circledast \mathbf{x}_t$ respectively capture the update from the hidden states at the previous step and the new input. The tensor-dot operation $\circledast$ is defined as the product of two tensors along the $N$ axis, e.g., $\mathcal{W}_j \circledast \tilde{\mathbf{h}}_{t-1} = [\mathbf{W}_j^1 \mathbf{h}_{t-1}^1, \cdots, \mathbf{W}_j^N \mathbf{h}_{t-1}^N]^\top$ where $\mathbf{W}_j^n \mathbf{h}_{t-1}^n \in \mathbb{R}^d$.

Depending on different update schemes of gates and memory cells, we proposed two realizations of IMV-LSTM, i.e. IMV-Full in Equation set 1 and IMV-Tensor in Equation set 2. In these two sets of equations, vec($\cdot$) refers to the vectorization operation, which concatenates columns of a matrix into a vector. $\oplus$ is the concatenation operation. $\odot$ denotes element-wise multiplication. In this paper, matricization($\cdot$) reshapes a vector of $\mathbb{R}^D$ into a matrix of $\mathbb{R}^{N \times d}$.

$$\begin{bmatrix} \mathbf{i}_t \\ \mathbf{f}_t \\ \mathbf{o}_t \end{bmatrix} = \sigma \left( \mathbf{W} \left[ \mathbf{x}_t \oplus \mathrm{vec}(\tilde{\mathbf{h}}_{t-1}) \right] + \mathbf{b} \right) \quad (2)$$

$$\mathbf{c}_t = \mathbf{f}_t \odot \mathbf{c}_{t-1} + \mathbf{i}_t \odot \mathrm{vec}(\tilde{\mathbf{j}}_t) \quad (3)$$

$$\tilde{\mathbf{h}}_t = \mathrm{matricization}(\mathbf{o}_t \odot \tanh(\mathbf{c}_t)) \quad (4)$$

Equation set 1: IMV-Full

$$\begin{bmatrix} \tilde{\mathbf{i}}_t \\ \tilde{\mathbf{f}}_t \\ \tilde{\mathbf{o}}_t \end{bmatrix} = \sigma \left( \mathcal{W} \circledast \tilde{\mathbf{h}}_{t-1} + \mathcal{U} \circledast \mathbf{x}_t + \mathbf{b} \right) \quad (5)$$

$$\tilde{\mathbf{c}}_t = \tilde{\mathbf{f}}_t \odot \tilde{\mathbf{c}}_{t-1} + \tilde{\mathbf{i}}_t \odot \tilde{\mathbf{j}}_t \quad (6)$$

$$\tilde{\mathbf{h}}_t = \tilde{\mathbf{o}}_t \odot \tanh(\tilde{\mathbf{c}}_t) \quad (7)$$

Equation set 2: IMV-Tensor

**IMV-Full**: With vectorization in Eq. (2) and (3), IMV-Full updates gates and memories using full $\tilde{\mathbf{h}}_{t-1}$ and $\tilde{\mathbf{j}}_t$ regardless of the variable-wise data in them. By simple replacement of the hidden update in standard LSTM by $\tilde{\mathbf{j}}_t$, IMV-Full behaves identically to standard LSTM while enjoying the interpretability shown below.

**IMV-Tensor**: By applying tensor-dot operations in Eq. (5), gates and memory cells are matrices as well, elements of which have the correspondence to input variables as hidden state matrix $\tilde{\mathbf{h}}_t$ does. $\mathcal{W}$ and $\mathcal{U}$ have the same shapes as $\mathcal{W}_j$ and $\mathcal{U}_j$ in Eq. (1)

In IMV-Full and IMV-Tensor, gates only scale $\tilde{\mathbf{j}}_t$ and $\tilde{\mathbf{c}}_{t-1}$ and thus retain the variable-wise data organization in $\tilde{\mathbf{h}}_t$. Meanwhile, based on tensorized hidden state Eq. (1) and gate update Eq. (5), IMV-Tensor can also be considered as a set of parallel LSTMs, each of which processes one variable series. The derived hidden states specific to individual variables are aggregated on both temporal and variable level through the attention described below.

## 3.2 MIXTURE ATTENTION

After feeding a sequence of $\{\mathbf{x}_1, \cdots, \mathbf{x}_T\}$ into either IMV-Full or IMV-Tensor, we obtain a sequence of hidden state matrices $\{\tilde{\mathbf{h}}_1, \cdots, \tilde{\mathbf{h}}_T\}$, where the sequence of hidden states specific to variable $n$ is extracted as $\{\mathbf{h}_1^n, \cdots, \mathbf{h}_T^n\}$.

In this part, we present the novel mixture attention mechanism in IMV-LSTM based on the following idea. Temporal attention is first applied to the sequence of hidden states corresponding to each variable, so as to obtain the summarized history of each variable. Then by using the history enriched hidden state of each variable, global variable attention is derived. These two steps are assembled into a probabilistic mixture model (Zong et al., 2018; Graves, 2013; Bishop, 1994), which facilitates the subsequent training, inference, and interpretation process.

In particular, the mixture attention is formulated as:

$$\begin{aligned} p(y_{T+1} \,|\, \mathbf{X}_T) &= \sum_{n=1}^{N} p(y_{T+1} | z_{T+1} = n, \mathbf{X}_T) \cdot p(z_{T+1} = n | \mathbf{X}_T) \\ &= \sum_{n=1}^{N} p(y_{T+1} \,|\, z_{T+1} = n, \mathbf{h}_1^n, \cdots, \mathbf{h}_T^n) \cdot p(z_{T+1} = n \,|\, \tilde{\mathbf{h}}_1, \cdots, \tilde{\mathbf{h}}_T) \\ &= \sum_{n=1}^{N} p(y_{T+1} \,|\, z_{T+1} = n, \underbrace{\mathbf{h}_T^n \oplus \mathbf{g}^n}_{\substack{\text{variable-wise} \\ \text{temporal attention}}}) \cdot \underbrace{p(z_{T+1} = n \,|\, \mathbf{h}_T^1 \oplus \mathbf{g}^1, \cdots, \mathbf{h}_T^N \oplus \mathbf{g}^N)}_{\text{overall variable attention}} \end{aligned} \quad (8)$$

In Eq. (8), we introduce a latent random variable $z_{T+1}$ into the the density function of $y_{T+1}$ to govern the generation process. $z_{T+1}$ is a discrete variable over the set of values $\{1, \cdots, N\}$ corresponding to $N$ input variables. Mathematically, $p(y_{T+1} \,|\, z_{T+1} = n, \mathbf{h}_T^n \oplus \mathbf{g}^n)$ characterizes the density of $y_{T+1}$ conditioned on historical data of variable $n$, while the prior of $z_{T+1}$, i.e. $p(z_{T+1} = n \,|\, \mathbf{h}_T^1 \oplus \mathbf{g}^1, \cdots, \mathbf{h}_T^N \oplus \mathbf{g}^N)$ controls to what extent $y_{T+1}$ is driven by variable $n$.

The context vector $\mathbf{g}^n$ is computed as the temporal attention weighted sum of hidden states corresponding to variable $n$, i.e., $\mathbf{g}^n = \sum_t \alpha_t^n \mathbf{h}_t^n$ where attention weight $\alpha_t^n = \frac{\exp(\mathrm{f}_n(\mathbf{h}_t^n))}{\sum_k \exp(\mathrm{f}_n(\mathbf{h}_k^n))}$. $\mathrm{f}_n(\cdot)$ can be a flexible function specific to variable $n$, e.g., neural networks (Bahdanau et al., 2014). The $p(y_{T+1} \,|\, z_{T+1} = n, \mathbf{h}_T^n \oplus \mathbf{g}^n)$ is a Gaussian distribution parameterized by $[\mu_n, \sigma_n] = \varphi_n(\mathbf{h}_T^n \oplus \mathbf{g}^n)$, where $\varphi_n(\cdot)$ can be a feedforward neural network.

The overall variable attention $p(z_{T+1} = n \mid \mathbf{h}_T^1 \oplus \mathbf{g}^1, \cdots, \mathbf{h}_T^N \oplus \mathbf{g}^N)$ is derived by a softmax function over $\{\mathrm{f}(\mathbf{h}_T^n \oplus \mathbf{g}^n)\}_N$, where $\mathrm{f}(\cdot)$ can be a feedforward neural network shared by all variables.

### 3.3 Learning, Inference, and Interpretation

In the learning phase, the set of parameters in IMV-Full or IMV-Tensor as well as the attention functions is denoted by $\Theta$. Given a set of $M$ training sequences $\{\mathbf{X}_T\}_M$ and $\{y_{T+1}\}_M$, the loss function to optimize is defined based on the negative log likelihood of the mixture attention model in Eq. (8) plus regularization terms.

In the inference phase, the prediction of $y_{T+1}$ is obtained by the weighted sum of means as :
$\hat{y}_{T+1} = \sum_n \mu_n \cdot p(z_{T+1} = n \mid \mathbf{h}_T^1 \oplus \mathbf{g}^1, \cdots, \mathbf{h}_T^N \oplus \mathbf{g}^N)$.

Regarding interpretation, we first illustrate the burden of deciphering variable and temporal importance from the raw attentions mentioned above. For instance, during the training on PLANT dataset used in the experiment section, we collect variable-wise temporal attention and overall variable attention values w.r.t. each training instance at each epoch. In Fig 2, Panel (a) plots the histograms of overall variable attention of three variables in PLANT at two different epochs. Ideally, attention weights of different variables are expected to distribute distinctly. However, through histograms in Panel (a) it is nontrivial to fully discriminate variable importance. Likewise, in Panel (b) plotting the histogram of temporal attention at some time lags of variable "P-temperature" at two different epochs does not ease the importance interpretation. Time lag represents the look-back time step w.r..t the current one. Similar phenomena are observed in other variables and datasets during the experiments.

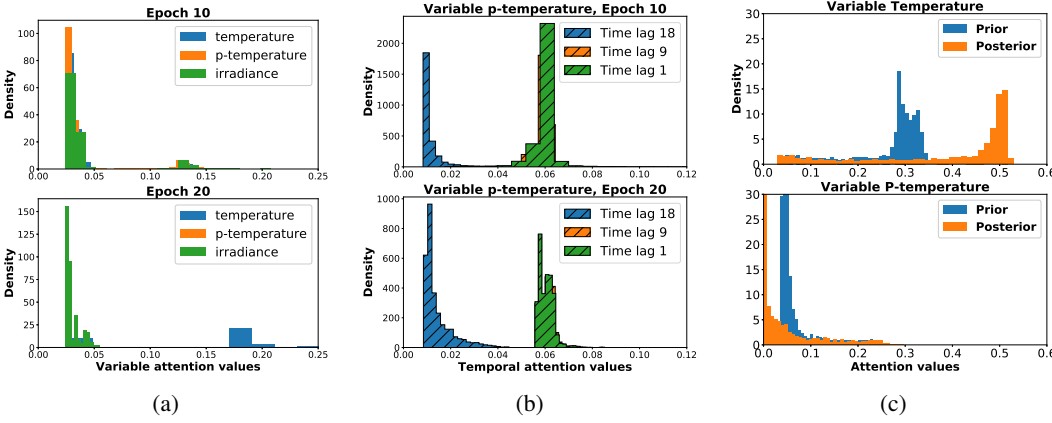

Figure 2: (a) Histograms of overall variable attention at different epochs. (b) Histograms of temporal attention of variable "P-temperature". (c) Prior and posterior attention histograms of two example variables.

Therefore, we proposed the following summarization method over temporal and variable attentions of data instances to quantify temporal and variable importance. First, we define the function $f_{agg}: \mathbb{R}^{A \times B} \to \mathbb{R}^B$, which maps an input $\mathbf{Z} \in \mathbb{R}^{A \times B}$ into an aggregated vector $\bar{\mathbf{z}} \in \mathbb{R}^B$. In the present paper, we choose the simple normalized summation function as:

$$\bar{\mathbf{z}} = f_{agg}(\mathbf{Z}) := \left[ \frac{\sum_a z_{a,1}}{\sum_a \sum_b z_{a,b}}, \cdots, \frac{\sum_a z_{a,B}}{\sum_a \sum_b z_{a,b}} \right]. \tag{9}$$

It is flexible to choose alternative functions for $f_{agg}$, e.g. robust statistics based methods, etc.

For temporal importance, we collect the temporal attention w.r.t. $n$-th variable of each instance $m$ as $\boldsymbol{\alpha}_m^n = [\alpha_{m,1}^n, \cdots, \alpha_{m,T-1}^n] \in \mathbb{R}^{T-1}$. Then, the unified temporal importance of variable $n$ is obtained as $\bar{\boldsymbol{\alpha}}^n = f_{agg}([\boldsymbol{\alpha}_1^n, \cdots, \boldsymbol{\alpha}_M^n]^\top)$, $\sum_t \bar{\alpha}_t^n = 1$, $\bar{\alpha}_t^n \in [0, 1]$.

The overall variable importance is formulated by using a novel posterior variable attention. Concretely, the posterior variable attention is derived as:

$$q^n := p(z_{T+1} = n \mid \mathbf{X}_T, y_{T+1}) \propto \mathcal{N}(y_{T+1} \mid \varphi_n(\mathbf{h}_T^n \oplus \mathbf{g}^n)) \frac{\exp(\mathrm{f}(\mathbf{h}_T^n \oplus \mathbf{g}^n))}{\sum_k \exp(\mathrm{f}(\mathbf{h}_T^k \oplus \mathbf{g}^k))}, \tag{10}$$

where $q^n$ provides more distinguishable attention distribution of different variables by taking the predictive likelihood into account. For instance, Panel (c) in Fig. (2) demonstrates the histograms of

posterior and prior attention (i.e. $p(z_{T+1} = n | \mathbf{X}_T)$) of two example variables in PLANT. Compared with priors, the posterior attentions of more important variable "Temperature" further shift rightward, while the posterior of less important variable "P-temperature" moves towards zero.

The unified variable importance can then be defined as $\bar{\boldsymbol{q}} = f_{agg}([\boldsymbol{q}_1, \cdots, \boldsymbol{q}_M]^\top)$, $\sum_n \bar{q}^n = 1$, $\bar{q}^n \in [0, 1]$, where $\boldsymbol{q}_m = [q_m^1, \cdots q_m^N] \in \mathbb{R}^N$ is the overall variable attention of instance $m$.

## 4 EXPERIMENTS

### 4.1 DATASETS

**NASDAQ** is the dataset from (Qin et al., 2017). It contains 81 major corporations under NASDAQ 100, as exogenous time series. The index value of the NASDAQ 100 is the target series. The frequency of the data collection is minute-by-minute. The first 35,100, the following 2,730 and the last 2,730 data points are respectively used as the training, validation and test sets.

**PLANT:** This dataset records the time series of energy production of a photo-voltaic power plant in Italy (Ceci et al., 2017). Exogenous data consists of 9 weather conditions variables (such as temperature, cloud coverage, etc.). The power production is the target. It provides 20842 sequences split into training (70%), validation (10%) and testing sets (20%).

**SML** is a public dataset used for indoor temperature forecasting. Same as (Qin et al., 2017), the room temperature is taken as the target series and another 16 time series are exogenous series. The data were sampled every minute. The first 3200, the following 400 and the last 537 data points are respectively used for training, validation, and test.

### 4.2 BASELINES AND EVALUATION SETUP

The first category of statistics baselines includes:

**STRX** is the structural time series model with exogenous variables (Scott & Varian, 2014; Radinsky et al., 2012). It is formulated in terms of unobserved components via the state space method.

**ARIMAX** is the auto-regressive integrated moving average with regression terms on exogenous variables (Hyndman & Athanasopoulos, 2014). It is a special case of vector auto-regression in this scenario.

The second category of machine learning baselines includes:

**RF** refers to random forests. It is an ensemble learning method consisting of several decision trees (Liaw et al., 2002; Meek et al., 2002) and was used in time series prediction (Patel et al., 2015).

**XGT** refers to the extreme gradient boosting (Chen & Guestrin, 2016). It is the application of boosting methods to regression trees (Friedman, 2001).

**ENET** represents Elastic-Net, which is a regularized regression method combining both L1 and L2 penalties of the lasso and ridge methods (Zou & Hastie, 2005) and has been used in time series analysis (Liu et al., 2010; Bai & Ng, 2008).

The third category of neural network baselines includes:

**RETAIN** uses RNNs to respectively learn weights on input data, which are then used to perform prediction (Choi et al., 2016).

**DUAL** is an encoder-decoder architecture, which uses an encoder LSTM to learn weights and then feeds pre-weighted input data into a decoder LSTM for forecasting (Qin et al., 2017).

In ARIMAX, the orders of auto-regression and moving-average terms are set via the autocorrelation and partial autocorrelation. For RF and XGT, the hyper-parameter tree depth and the number of iterations are chosen from range $[3, 10]$ and $[2, 200]$ via grid search. For XGT, L2 regularization is added by searching within $\{0.0001, 0.001, 0.01, 0.1, 1, 10\}$. As for ENET, the coefficients for L2 and L1 penalties are selected from $\{0, 0.1, 0.3, 0.5, 0.7, 0.9, 1, 1.5, 2\}$. For machine learning baselines, multi-variable input sequences are flattened into feature vectors.

We implemented IMV-LSTM and neural network baselines with Tensorflow[2]. We used Adam with the mini-batch of $64$ instances (Kingma & Ba, 2014). For the size of recurrent and dense layers in the baselines, we conduct grid search over $\{16, 32, 64, 128, 256, 512\}$. The size of the IMV-LSTM layer is set by the number of neurons per variable selected from $\{10, 15, 20, 25\}$. Dropout is selected in $\{0.8, 0.5\}$. Learning rate is searched in $\{0.0005, 0.001, 0.005, 0.01, 0.05\}$. L2 regularization is added with the coefficient chosen from $\{0.0001, 0.001, 0.01, 0.1, 1.0\}$. We train each approach 10 times and report average performance. For baseline DUAL on NASDAQ and SML datasets, we use the hyper-parameters achieving the best performance in Qin et al. (2017). On PLANT dataset, hyper-parameters are searched in above sets of values. The window size for NASDAQ and SML, namely $T$ in Sec. 3, is set to 10 according to Qin et al. (2017), while for PLANT it is 20 to test long temporal dependency.

We consider two metrics to measure the prediction performance. Specifically, RMSE is defined as $\text{RMSE} = \sqrt{\sum_m (y_m - \hat{y}_m)^2 / M}$. MAE is defined as $\text{MAE} = \sum_m |y_m - \hat{y}_m| / M$.

### 4.3 PREDICTION PERFORMANCE

We report the prediction errors in Table 1, each cell of which presents the average RMSE and MAE with standard errors. Note that IMV-Full and IMV-Tensor are single network structures. Their good prediction performance below verifies the idea that instead of complex network architecture in baselines, simple and proper mixture of well-maintained variable-wise hidden states also improves the prediction performance as well as empowering the interpretability shown below. In particular, IMV-LSTM family outperforms baselines by around $80\%$ at most. IMV-Full performs mostly better than baselines, while IMV-Tensor surpasses IMV-Full on NASDAQ and SML datasets.

Table 1: RMSE and MAE with std. errors

| Dataset | NASDAQ | PLANT | SML |
|---------|--------|-------|-----|
| STRX | $0.41 \pm 0.01, 0.35 \pm 0.02$ | $231.43 \pm 0.19, 193.23 \pm 0.43$ | $0.039 \pm 0.001, 0.033 \pm 0.001$ |
| ARIMAX | $0.34 \pm 0.02, 0.23 \pm 0.03$ | $225.54 \pm 0.23, 193.42 \pm 0.41$ | $0.060 \pm 0.002, 0.053 \pm 0.002$ |
| RF | $0.31 \pm 0.02, 0.27 \pm 0.03$ | $164.23 \pm 0.65, 130.90 \pm 0.15$ | $0.045 \pm 0.001, 0.032 \pm 0.001$ |
| XGT | $0.28 \pm 0.01, 0.23 \pm 0.02$ | $164.10 \pm 0.54, 131.47 \pm 0.21$ | $0.017 \pm 0.001, 0.013 \pm 0.001$ |
| ENET | $0.31 \pm 0.03, 0.21 \pm 0.01$ | $168.22 \pm 0.49, 137.04 \pm 0.38$ | $0.018 \pm 0.001, 0.015 \pm 0.001$ |
| DUAL | $0.31 \pm 0.003, 0.21 \pm 0.002$ | $163.29 \pm 0.54, 130.87 \pm 0.12$ | $0.019 \pm 0.001, 0.015 \pm 0.001$ |
| RETAIN | $0.12 \pm 0.07 , 0.11 \pm 0.06$ | $250.69 \pm 0.36, 190.11 \pm 0.15$ | $0.048 \pm 0.001, 0.037 \pm 0.001$ |
| IMV-Full | $0.27 \pm 0.01, 0.23 \pm 0.01$ | $\mathbf{157.23 \pm 0.16, 128.13 \pm 0.14}$ | $0.015 \pm 0.002, 0.012 \pm 0.001$ |
| IMV-Tensor | $\mathbf{0.09 \pm 0.005, 0.07 \pm 0.004}$ | $159.90 \pm 0.22, 129.43 \pm 0.10$ | $\mathbf{0.009 \pm 0.0009, 0.006 \pm 0.0005}$ |

### 4.4 INTERPRETATION

In this part, we investigate the interpretability of IMV-Full and IMV-Tensor by collecting the variable and temporal importance values during the training under the best hyper-parameters. As far as we know, experiments in previous work using attention in RNNs do not unveil such fine-grained interpretation over both variable and temporal level. Due to the page limitation, the interpretability results on other datasets are in the appendix section.

In Fig. 3, Panel (a) shows the overall variable importance values w.r.t. training epochs on PLANT dataset. Specifically, as variable importance converges, the ranking of variables is clearly identified at the end of the training, i.e. variables with high importance values are top ranked. Meanwhile, Panel (b) demonstrates the temporal importance values of each variable at the beginning and ending of the training (i.e. epoch 0 and 75). The lighter the color, the higher the temporal importance value of the corresponding time lag. At epoch 0, the randomly initialized network gives rises to similar temporal importance pattern for most of the variables. At epoch 75, diverse patterns w.r.t different variables are learned. For instance, the importance value at around time lag 1 to 4 of variable "P-temperature" are obviously higher, which indicates that this variable has instant relation to the target. For variable "wind-speed", this heat map tells that its historical observations after time lag 14 could be negligibly correlated to the target.

Panel (c) shows the convergence of the variable importance of IMV-Tensor on PLANT dataset. The ranking of variable importance is slightly different to that in Panel (a). It is because in IMV-Tensor the gate and memory update scheme makes hidden states w.r.t. different variables evolve independently,

---

[2] Code will be released upon requested.

thereby leading to different hidden states and attention values to IMV-Full. However, we can still decipher something in common. For instance, variable "wind-speed" stays relatively important in both Panel (a) and (c), i.e. rank 1st and 4th respectively. As for temporal importance, in Panel (d) variable "P-temperature" presents temporal importance pattern similar to that in Panel (b). It is also worth exploring whether we can obtain more consistent variable and temporal importance from IMV-Full and IMV-Tensor in the future work.

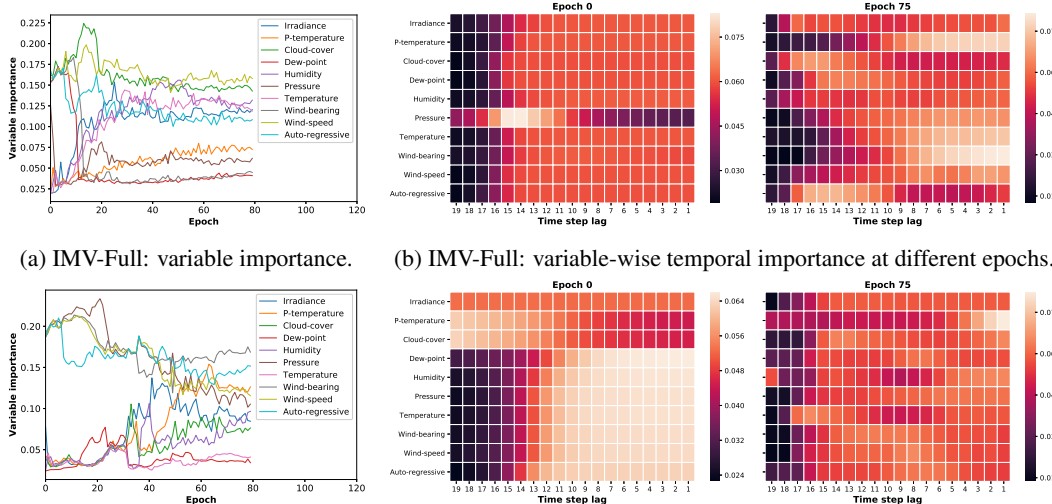

(a) IMV-Full: variable importance.    (b) IMV-Full: variable-wise temporal importance at different epochs.

(c) IMV-Tensor: variable importance.    (d) IMV-Tensor: variable-wise temporal importance at different epochs.

Figure 3: Variable and temporal importance interpretation during the training of IMV-Full and IMV-Tensor on PLANT dataset. (Best viewed in color)

### 4.5 VARIABLE IMPORTANCE FOR PREDICTION

In this group of experiments, we evaluate the efficacy of variable importance through the lens of prediction tasks. We focus on IMV-LSTM family and RNN baselines, i.e. DUAL and RETAIN. Specifically, for each approach, we first rank variables respectively according to the variable importance in IMV-LSTM and variable attention in DUAL and RETAIN. Then we rebuild datasets only consisting of top 50% ranked variables for each approach (i.e. high importance or attention values) to retrain each model and obtain the prediction errors in Table 2. (The full ranking of variables is in the appendix.)

Ideally, effective variable importance leads to top variables highly related to the target and thus retrained models have comparable errors in comparison to their counterparts in Table 1. In particular, IMV-Full and IMV-Tensor present comparable and even lower errors in Table 2, while DUAL and RETAIN have higher errors mostly. An additional advantage of using top variables is the training efficiency, e.g. the training time of each epoch in IMV-Tensor is reduced from ~16 sec to ~11 sec.

Table 2: RMSE and MAE with std. errors under top 50% important variables

| Dataset | NASDAQ | PLANT | SML |
|---|---|---|---|
| DUAL | $0.16 \pm 0.08, 0.16 \pm 0.05$ | $171.30 \pm 0.17, 154.15 \pm 0.20$ | $0.026 \pm 0.002, 0.018 \pm 0.002$ |
| RETAIN | $0.17 \pm 0.03, 0.15 \pm 0.02$ | $226.38 \pm 0.72, 167.90 \pm 0.81$ | $0.060 \pm 0.001, 0.044 \pm 0.004$ |
| IMV-Full | $0.26 \pm 0.01, 0.23 \pm 0.02$ | $162.14 \pm 0.10, \mathbf{128.51 \pm 0.12}$ | $0.015 \pm 0.001, 0.011 \pm 0.002$ |
| IMV-Tensor | $\mathbf{0.12 \pm 0.007, 0.10 \pm 0.01}$ | $\mathbf{157.64 \pm 0.14}, 128.86 \pm 0.13$ | $\mathbf{0.007 \pm 0.0005, 0.006 \pm 0.0003}$ |

## 5 CONCLUSION

In this paper, we propose an interpretable multi-variable LSTM (IMV-LSTM) for time series with exogenous variables. Based on the hidden state matrix and update scheme, we present two realizations i.e. IMV-Full and IMV-Tensor as well as developing mixture temporal and variable attention mechanism. It enables to infer and quantify fine-grained variable-wise temporal importance and overall variable importance w.r.t. the target. Extensive experiments exhibit the superior prediction performance, interpretability and efficacy of variable importance of IMV-LSTM.

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

## 6 APPENDIX

### 6.1 INTERPRETABLE MULTI-VARIABLE LSTM

In IMV-Full and IMV-Tensor, the learning phase aims to minimize the loss function as follows:

$$\mathcal{L}(\Theta) = -\sum_{m=1}^{M} \log \sum_{n=1}^{N} \mathcal{N}(y_{T+1,m} \,|\, \varphi_n(\,\mathbf{h}_{T,m}^n \oplus \mathbf{g}_m^n\,)) \frac{\exp\left(\mathrm{f}(\,\mathbf{h}_{T,m}^n \oplus \mathbf{g}_m^n\,)\right)}{\sum_k \exp\left(\mathrm{f}(\,\mathbf{h}_{T,m}^k \oplus \mathbf{g}_m^k\,)\right)} + \lambda \|\Theta\|^2 \quad (11)$$

The derivation of the posterior attention is as follows. This equation is based on the Bayesian rule to derive the posterior of variable attention, by taking into account both the predictive likelihood w.r.t. the corresponding variable and prior attention.

$$
\begin{aligned}
q^n &:= p(z_{T+1} = n \,|\, \mathbf{X}_T, y_{T+1}) \\
&\propto p(y_{T+1}|z_{T+1} = n, \mathbf{X}_T) \cdot p(z_{T+1} = n|\mathbf{X}_T) \\
&\approx p(y_{T+1} \,|\, z_{T+1} = n, \mathbf{h}_T^n \oplus \mathbf{g}^n) \cdot p(z_{T+1} = n \,|\, \mathbf{h}_T^1 \oplus \mathbf{g}^1, \cdots, \mathbf{h}_T^N \oplus \mathbf{g}^N) \quad (12) \\
&= \mathcal{N}(y_{T+1} \,|\, \varphi_n(\,\mathbf{h}_T^n \oplus \mathbf{g}^n\,)) \frac{\exp\left(\mathrm{f}(\,\mathbf{h}_T^n \oplus \mathbf{g}^n\,)\right)}{\sum_k \exp\left(\mathrm{f}(\,\mathbf{h}_T^k \oplus \mathbf{g}^k\,)\right)} ,
\end{aligned}
$$

where the $p(y_{T+1} \,|\, z_{T+1} = n, \mathbf{h}_T^n \oplus \mathbf{g}^n)$ is a Gaussian distribution parameterized by $[\,\mu_n, \sigma_n] = \varphi_n(\,\mathbf{h}_T^n \oplus \mathbf{g}^n\,)$, where $\varphi_n(\cdot)$ can be a feedforward neural network.

Regarding the summarization of variable-wise temporal attention and variable attention of each instance, since each attention is a sample of discrete distribution (i.e. defined on temporal steps and input variables), alternative summarization method is to fit a Dirichlet distribution to these attentions and use the estimated expectation as the importance values.

Next, we provide the formal analysis about the complexity of IMV-LSTM through Lemma 6.1 and Lemma 6.2.

**Lemma 6.1.** *Given time series of $N$ variables, assume a standard LSTM and IMV-LSTM layer both have size $D$, i.e. $D$ neurons in the layer. Then, compared to the number of parameters of the standard LSTM, IMV-Full and IMV-Tensor respectively reduce the network complexity by $(N-1)D + (1 - 1/N)D \cdot D$ and $4(N-1)D + 4(1 - 1/N)D \cdot D$ number of parameters.*

*Proof.* In a standard LSTM of layer size $D$, trainable parameters lie in the hidden and gate update functions. In total, these update functions have $4D \cdot D + 4N \cdot D + 4D$ parameters, where $4D \cdot D + 4N \cdot D$ comes from the transition and $4D$ corresponds to the bias terms.

For IMV-Full, assume each input variable corresponds to one-dimensional time series. Based on Eq. 1, the hidden update has $2D + D^2/N$ trainable parameters. Equation set 1 gives rise to the number of parameters equal to that of the standard LSTM. Therefore, the reduce number of parameters is $(N-1)D + (1-1/N)D \cdot D$. As for IMV-Tensor, more parameter reduction stems from that the gate update functions in Equation set 2 make use of the tensor-dot operation as Eq. 1. Likewise, the reduced amount of parameters can be derived. $\square$

**Lemma 6.2.** *For time series of $N$ variables and the recurrent layer of size $D$, IMV-Full and IMV-Tensor respectively have the computation complexity at each update step as: $\mathcal{O}(D^2 + N \cdot D)$ and $\mathcal{O}(D^2/N + D)$.*

*Proof.* Assume that $D$ neurons of the recurrent layer in IMV-Full and IMV-Tensor are evenly assigned to $N$ input variables, namely each input variable has $d = D/N$ corresponding neurons. For IMV-Full, based on Eq. 1, the hidden update has computation complexity $N \cdot d^2 + N \cdot d$, while the gate update process has the complexity $D^2 + N \cdot D$. Overall, the computation complexity is $\mathcal{O}(D^2 + N \cdot D)$, which is identical to the complexity of a standard LSTM.

As for IMV-Tensor, since the gate update functions in Equation set 2 make use of the tensor-dot operation as Eq. 1, gate update functions have the same computation complexity as Eq. 1. The overall complexity is $\mathcal{O}(D^2/N + D)$, which is $1/N$ of the complexity of a standard LSTM. $\square$

Basically, Lemma 6.1 and Lemma 6.2 indicate that a high number of input variables leads to a large portion of parameter and computation reduction in IMV-LSTM family.

## 6.2 EXPERIMENTS

In this part, we provide complementary experiment results as well as the insights from the results.

### 6.2.1 PREDICTION PERFORMANCE ANALYSIS

**Effect of exogenous variables**   The first new set of experiments reports the prediction errors of statistical, machine learning models and LSTM with attention trained with only auto-regressive data of the target variable. By comparing with the results of the models using both the target and exogenous variables, it is aimed to demonstrate that adding exogenous variables benefits the prediction performance.

This group of new results is put together with Table 1 into Table 3 below. For uni-variable input, DUAL, RETAIN and our IMV-LSTM reduce to a standard LSTM with temporal attention, and thus we report the results under the name of LSTM-Att in Table 3. In Table 3, it is observed that the models trained with both target and exogenous variables obtain lower errors. IMV-Tensor mostly outperforms IMV-Full. Though IMV-Full performs a little better than IMV-Tensor on PLANT dataset, they are quite comparable. IMV-Tensor has a much lower error (e.g. $50\%$ less at the maximum) for the rest of experiments.

**Effect of the target variable**   Meanwhile, we add another group of experiments to evaluate the performance of IMV-LSTM without the target variable. For the ease of comparison, the prediction errors of IMV-LSTM on data including and excluding the target variable are put together in Table 4.

In comparison to using the target variable, IMV-LSTM family presents comparable prediction performance without the target variable, though the errors arise a little. Error increases by around $20\%$ without using the target variable. It suggests that in the experiment datasets, overall exogenous variables are correlated to the target and contribute to the prediction. Auto-regressive history also has the certain correlation to the future of the target variable.

**Insights from the results**   The superior experimental performance of IMV-Tensor demonstrates a meaningful phenomenon. For multi-variable data, properly modeling individual variables and their interaction in LSTM is not only important for the performance, but also brings additional benefits, i.e. the interpretability in this paper.

Specifically, when using LSTM on multi-variable data, the conventional way is to directly feed multi-variable data into LSTM for target prediction. IMV-Full decomposes the hidden states, as well as keeping the variable interaction in the gate updating. IMV-Tensor goes one step further. It amounts to fit parallel LSTMs into the probabilistic mixture framework. It allows to first model individual variables and then collectively perform the prediction via the mixture attention. As a result, it enjoys theoretical soundness, even better prediction performance as well as interpretability.

Table 3: RMSE and MAE with std. errors

| Dataset | NASDAQ | PLANT | SML |
|---|---|---|---|
| Only auto-regressive data of the target variable | | | |
| STR | $0.47 \pm 0.01, 0.40 \pm 0.01$ | $243.35 \pm 0.52, 210.31 \pm 0.35$ | $0.048 \pm 0.001, 0.041 \pm 0.002$ |
| ARIMA | $0.39 \pm 0.01, 0.31 \pm 0.02$ | $231.65 \pm 0.42, 205.24 \pm 0.64$ | $0.073 \pm 0.003, 0.065 \pm 0.002$ |
| RF | $0.62 \pm 0.03, 0.57 \pm 0.02$ | $168.02 \pm 0.23, 135.20 \pm 0.15$ | $0.048 \pm 0.001, 0.035 \pm 0.001$ |
| XGT | $0.61 \pm 0.02, 0.57 \pm 0.02$ | $165.17 \pm 0.35, 134.71 \pm 0.21$ | $0.024 \pm 0.001, 0.020 \pm 0.001$ |
| ENET | $0.35 \pm 0.02, 0.22 \pm 0.01$ | $169.37 \pm 0.42, 143.45 \pm 0.38$ | $0.023 \pm 0.001, 0.019 \pm 0.001$ |
| LSTM-Att | $0.41 \pm 0.01, 0.34 \pm 0.01$ | $165.98 \pm 0.54, 135.23 \pm 0.12$ | $0.023 \pm 0.001, 0.020 \pm 0.002$ |
| Both target and exogenous variables | | | |
| STRX | $0.41 \pm 0.01, 0.35 \pm 0.02$ | $231.43 \pm 0.19, 193.23 \pm 0.43$ | $0.039 \pm 0.001, 0.033 \pm 0.001$ |
| ARIMAX | $0.34 \pm 0.02, 0.23 \pm 0.03$ | $225.54 \pm 0.23, 193.42 \pm 0.41$ | $0.060 \pm 0.002, 0.053 \pm 0.002$ |
| RF | $0.31 \pm 0.02, 0.27 \pm 0.03$ | $164.23 \pm 0.65, 130.90 \pm 0.15$ | $0.045 \pm 0.001, 0.032 \pm 0.001$ |
| XGT | $0.28 \pm 0.01, 0.23 \pm 0.02$ | $164.10 \pm 0.54, 131.47 \pm 0.21$ | $0.017 \pm 0.001, 0.013 \pm 0.001$ |
| ENET | $0.31 \pm 0.03, 0.21 \pm 0.01$ | $168.22 \pm 0.49, 137.04 \pm 0.38$ | $0.018 \pm 0.001, 0.015 \pm 0.001$ |
| DUAL | $0.31 \pm 0.003, 0.21 \pm 0.002$ | $163.29 \pm 0.54, 130.87 \pm 0.12$ | $0.019 \pm 0.001, 0.015 \pm 0.001$ |
| RETAIN | $0.12 \pm 0.07 , 0.11 \pm 0.06$ | $250.69 \pm 0.36, 190.11 \pm 0.15$ | $0.048 \pm 0.001, 0.037 \pm 0.001$ |
| IMV-Full | $0.27 \pm 0.01, 0.23 \pm 0.01$ | $\mathbf{157.23 \pm 0.16, 128.13 \pm 0.14}$ | $0.015 \pm 0.002, 0.012 \pm 0.001$ |
| IMV-Tensor | $\mathbf{0.09 \pm 0.005, 0.07 \pm 0.004}$ | $159.90 \pm 0.22, 129.43 \pm 0.10$ | $\mathbf{0.009 \pm 0.0009, 0.006 \pm 0.0005}$ |

Table 4: RMSE and MAE with std. errors

| Dataset | NASDAQ | PLANT | SML |
|---|---|---|---|
| Without the target variable | | | |
| IMV-Full | $0.32 \pm 0.01, 0.29 \pm 0.01$ | $163.47 \pm 0.21, 135.53 \pm 0.12$ | $0.019 \pm 0.001, 0.015 \pm 0.002$ |
| IMV-Tensor | $0.11 \pm 0.01, 0.09 \pm 0.01$ | $162.10 \pm 0.24, 134.99 \pm 0.14$ | $0.011 \pm 0.001, 0.010 \pm 0.002$ |
| Both target and exogenous variables | | | |
| IMV-Full | $0.27 \pm 0.01, 0.23 \pm 0.01$ | $\mathbf{157.23 \pm 0.16, 128.13 \pm 0.14}$ | $0.015 \pm 0.002, 0.012 \pm 0.001$ |
| IMV-Tensor | $\mathbf{0.09 \pm 0.005, 0.07 \pm 0.004}$ | $159.90 \pm 0.22, 129.43 \pm 0.10$ | $\mathbf{0.009 \pm 0.0009, 0.006 \pm 0.0005}$ |

### 6.2.2 VISUALIZATION OF PREDICTION ERRORS

In this part, the prediction testing errors of IMV-LSTM family are visualized.

Fig. 4 shows the Q-Q plot of testing errors on each dataset. Fig. 5 demonstrates the testing error curve during the training phase. Basically, we can observe that IMV-Tensor has lower prediction errors as well as more smooth testing error curves.

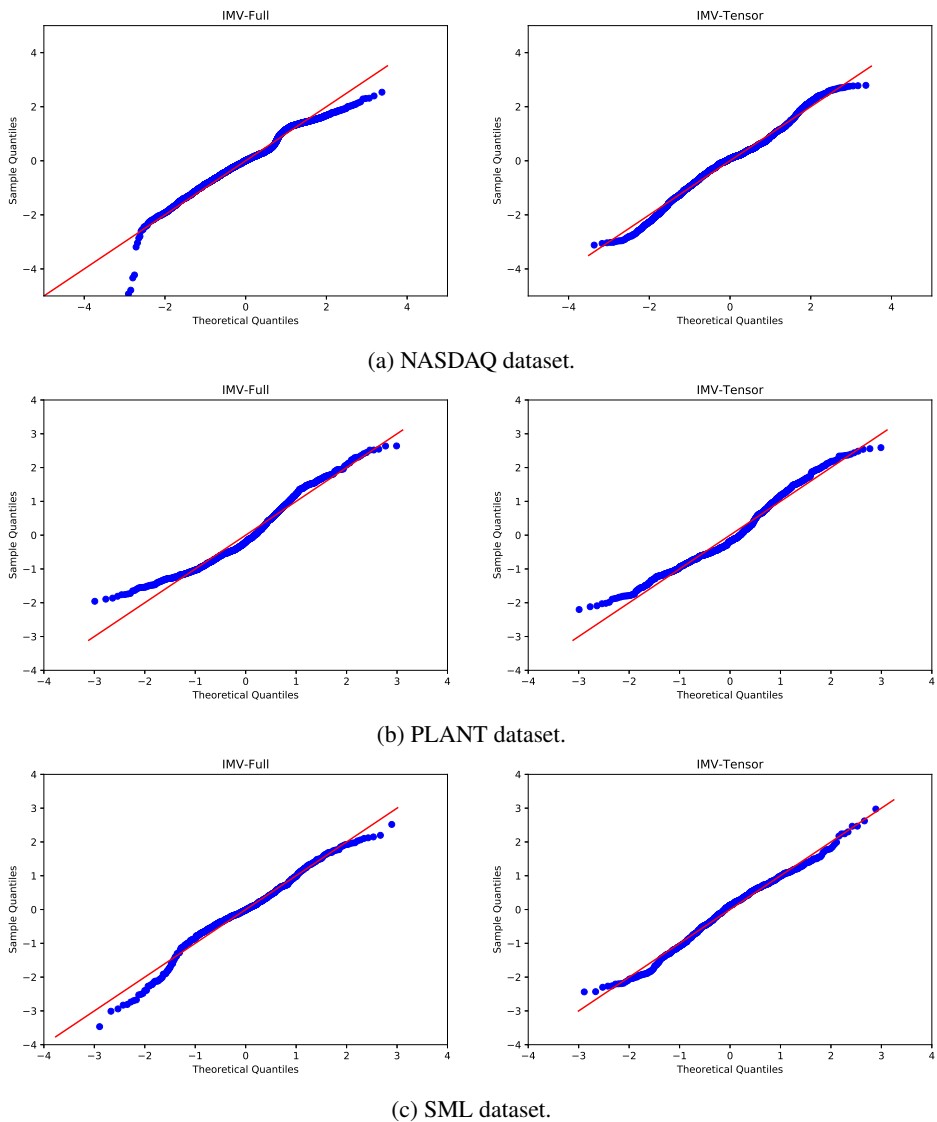

Figure 4: Q-Q plot of testing errors

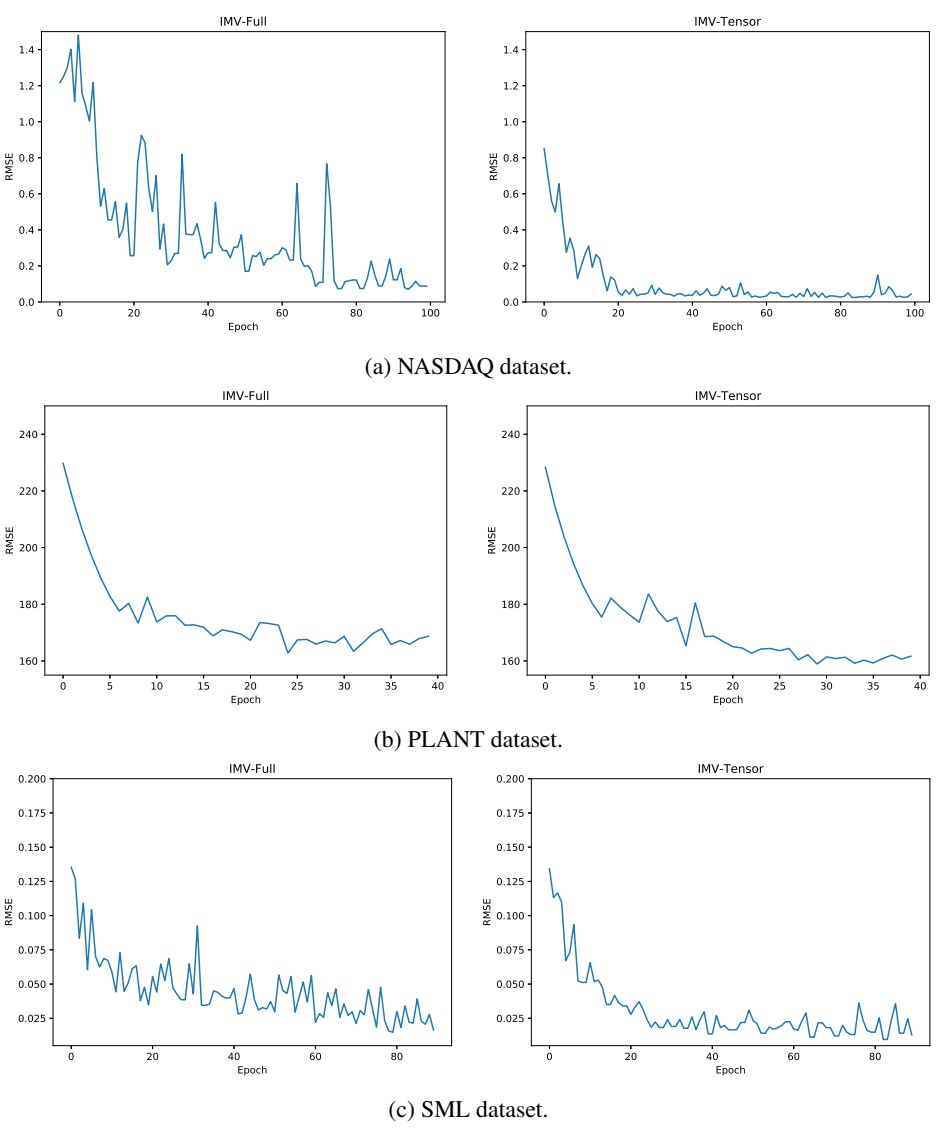

(a) NASDAQ dataset.

(b) PLANT dataset.

(c) SML dataset.

Figure 5: Testing error curves over the training phase

### 6.2.3 MODEL INTERPRETATION

In this part, we exploit the domain knowledge in the literature corresponding to the dataset, so as to qualitatively explain the importance learned by IMV-LSTM. Fig. 6 − 9 show the quantitative variable-wise temporal and variable importance values of IMV-LSTM on different datasets. Note that not all datasets in the experiments have associated study of variable importance in the literature. PLANT dataset has some related work on analyzing effective factors on energy production, therefore we will focus on explaining PLANT dataset with domain knowledge.

For PLANT dataset, as is discussed in (Mekhilef et al., 2012; Ghazi & Ip, 2014), wind and humidity affect the efficiency of photovoltaic cells and they are relatively high ranked by our variable importance. Humidity causes dust deposition and settlement and consequentially degradation in solar cell efficiency. Increased wind can move more heat from the PV cell surface as well as lowering the humidity of the atmospheric air in the surroundings, which leads to better efficiency. Also we obervat

For SML dataset, IMV-Tensor and IMV-Full share some variable importance in common. For instance, for the experiments in Sec. 4.5 on the SML dataset, in Table 7 they share 5 variables in common among the top 50% variables (i.e. 8 variables) of each method. Some of the different variables in top 50% refer to the similar concept, e.g. CO2 dining and CO2 room.

Considering the superior prediction performance of IMV-Tensor shown in Table 3 and Table 2, IMV-Tensor's importance values are favored. However, the difference between the importance learned by IMV-Full and IMV-Tensor still needs further investigation both theoretically and experimentally in the future work.

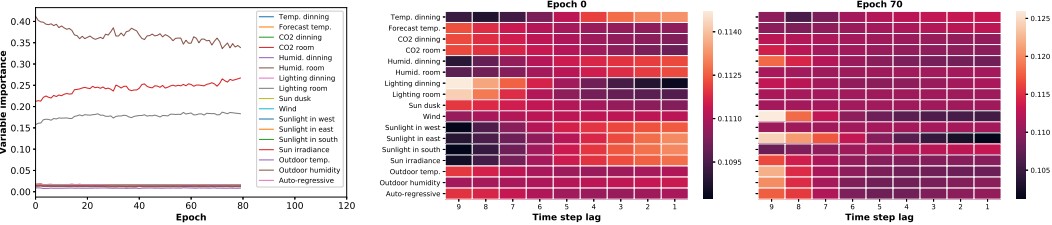

(a) Variable importance w.r.t. epochs.  (b) Variable-wise temporal importance at different epochs.

Figure 6: IMV-Full on SML dataset. (Best viewed in color)

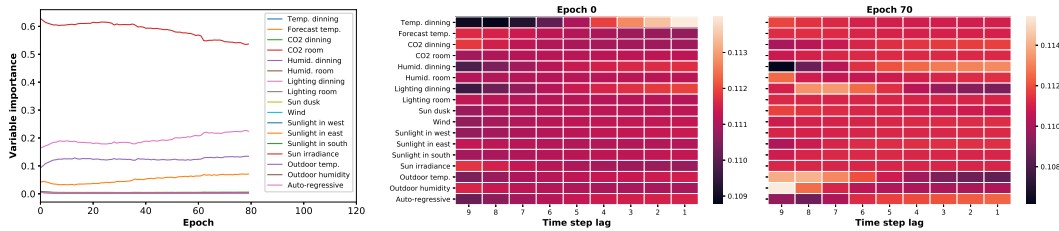

(a) Variable importance w.r.t. epochs.  (b) Variable-wise temporal importance at different epochs.

Figure 7: IMV-Tensor on SML dataset. (Best viewed in color)

In the following Table 5, 6, and 7, we list the full ranking of variables of the datasets by each approach. Variables associated with the importance or attention values are ranked in decreasing order.

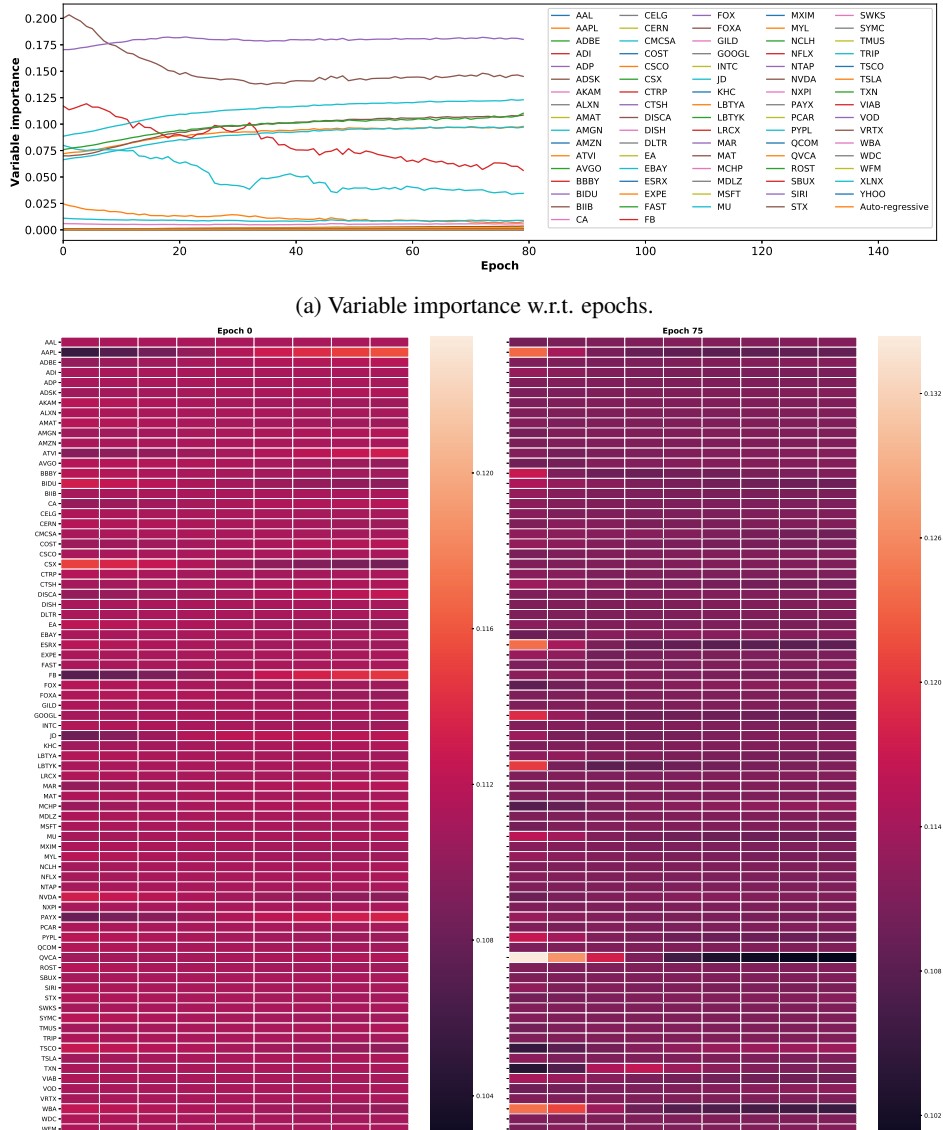

(a) Variable importance w.r.t. epochs.

(b) Variable-wise temporal importance at different epochs.

Figure 8: IMV-Full on NASDAQ dataset. (Best viewed in color)

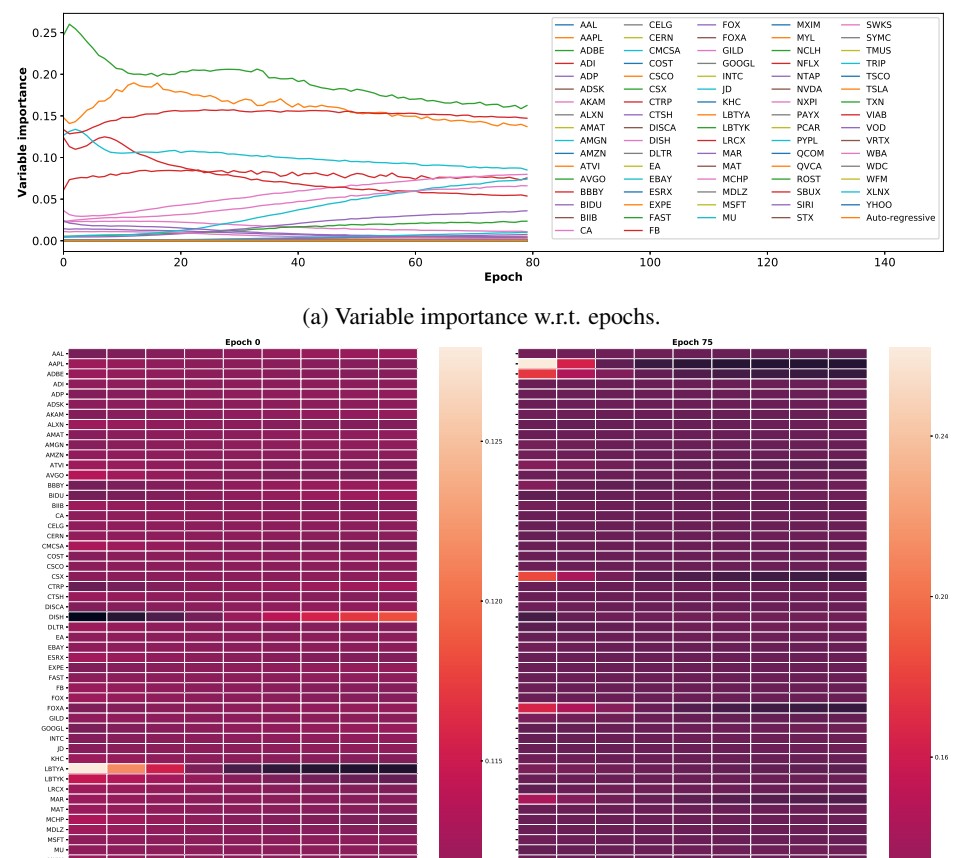

(a) Variable importance w.r.t. epochs.

(b) Variable-wise temporal importance at different epochs.

Figure 9: IMV-Tensor on NASDAQ dataset. (Best viewed in color)

Table 5: Variable importance ranking by IMV-Full and IMV-Tensor on NASDAQ dataset.

| Dataset | Method | Rank of variables according to importance |
|---|---|---|
| NASDAQ | IMV-Full | 'ADSK', 0.00023858716, 'PAYX', 0.00023869322, 'AAL', 0.00023993119, 'MYL', 0.00024015515, 'CA', 0.00024144033], 'FOX', 0.00024341498, 'EA', 0.00024963205], 'BIDU', 0.00025009923, 'MCHP', 0.00025015706, 'QVCA', 0.00025018162, 'NVDA', 0.00025088928, 'WBA', 0.00025147066, 'LRCX', 0.00025165512, 'TSCO', 0.00025247637, 'CTSH', 0.00025284023, 'CSX', 0.00025417344, 'COST', 0.00025498777, 'BIIB', 0.00025547648, 'LBTYA', 0.00025680827, 'SIRI', 0.00025686354, 'ADBE', 0.00025687047, 'MDLZ', 0.00025788756, 'LBTYK', 0.00025885308, 'INTC', 0.00025894548, 'TSLA', 0.0002592771, 'WFM', 0.00025941888, 'SBUX', 0.00025953245, 'AVGO', 0.00026012328], 'CTRP', 0.00026024296, 'AMZN', 0.00026168497, 'ALXN', 0.00026173133, 'AMGN', 0.0002617908, 'GILD', 0.0002619058, 'VOD', 0.00026195042, 'ROST', 0.00026237246, 'NXPI', 0.0002624988, 'KHC', 0.0002625609, 'ADP', 0.0002626155, 'WDC', 0.00026269013, 'QCOM', 0.00026288, 'TMUS', 0.00026333777, 'AMAT', 0.00026334616, 'AKAM', 0.00026453246, 'PCAR', 0.00026510606, 'CERN', 0.00026535543, 'VRTX', 0.00026579297, 'MU', 0.00026719182, 'MAR', 0.00026789604, 'TXN', 0.00026821258, 'GOOGL', 0.0002684545, 'ESRX', 0.00026995668, 'ATVI', 0.0002703378, 'STX', 0.0002708045, 'FAST', 0.00027182887, 'EXPE', 0.0002747627, 'CELG', 0.00027897576, 'PYPL', 0.00027971127, 'MXIM', 0.0002802631, 'NFLX', 0.00028330996, 'BBBY', 0.00028975168, 'SYMC', 0.0002932911, 'CMCSA', 0.00031882498, 'SWKS', 0.00034903747, 'DLTR', 0.0004099159, 'YHOO', 0.0004359138, 'VIAB', 0.00046212596, 'Auto-regressive', 0.0004718905, 'MAT', 0.0008193875, 'MSFT', 0.002350653, 'ADI', 0.0035426863, 'DISH', 0.0056709386, 'AAPL', 0.007597621, 'EBAY', 0.008922806, 'JD', 0.03449823, 'FB', 0.056254942, 'XLNX', 0.09711476, 'CSCO', 0.09782402, 'DISCA', 0.108503476, 'NCLH', 0.11029968, u'TRIP', 0.12302372, 'FOXA', 0.14510903, 'NTAP', 0.18010232 |
| | IMV-Tensor | 'ATVI', 0.00012247293, 'ADSK', 0.00012340973, 'FAST', 0.0001275845, 'WFM', 0.00013183481, 'ALXN', 0.00014380908, 'NFLX', 0.00014429294, 'QVCA', 0.00014494512, 'MSFT', 0.00014505234, 'BIDU', 0.00014950531, 'ESRX', 0.00015155961, 'DISCA', 0.00015276023, 'GILD', 0.00015325642], 'KHC', 0.00015800942, 'EBAY', 0.00015860644, 'NTAP', 0.00015893515, 'INTC', 0.0001592579, 'LBTYK', 0.00015955475], 'SWKS', 0.00015960005, 'SBUX', 0.0001602487, 'AMGN', 0.00016195989, 'AVGO', 0.00016398374, 'AMAT', 0.00016628107, 'FB', 0.0001681524], 'MYL', 0.00016860824, 'CELG', 0.00016944246, 'BIIB', 0.00016954532, 'CTRP', 0.00016966274, 'DLTR', 0.00017032732, 'ROST', 0.00017111507, 'MXIM', 0.00017283233, 'CTSH', 0.00017294307, 'TMUS', 0.00017294812], 'CERN', 0.00017299024, 'MDLZ', 0.0001731659, 'EA', 0.00017319905, 'CA', 0.00017323176, 'NVDA', 0.0001732874], 'COST', 0.00017329257, 'FOX', 0.00017335685, 'EXPE', 0.00017337044, 'CMCSA', 0.00017339012, 'QCOM', 0.00017341232, 'PCAR', 0.00017345154, 'ADI', 0.00017346471, 'TXN', 0.00017350669, 'PAYX', 0.00017363002, 'SYMC', 0.00017364287, 'TSCO', 0.0001738104, 'CSCO', 0.000173811, 'GOOGL', 0.0001748285, 'AMZN', 0.00018132143, 'STX', 0.00018217335, 'VRTX', 0.0001833355, 'MAT', 0.00018341257, 'AAL', 0.00019685448, 'YHOO', 0.00019987396, 'JD', 0.00020849655, 'XLNX', 0.0002687659, 'FOXA', 0.0004502886, 'WBA', 0.0004633159, 'Auto-regressive', 0.000513615, 'WDC', 0.00055753137, 'ADBE', 0.000716344, 'TSLA', 0.001061617, [u'MCHP', 0.0031253153, 'AAPL', 0.004319694, 'SIRI', 0.0043806615, 'VOD', 0.007254429, 'ADP', 0.012473345, 'AKAM', 0.012664658, 'TRIP', 0.016278176, 'MAR', 0.022737814, 'CSX', 0.028656403, 'MU', 0.05227967, 'BBBY', 0.07014921, 'DISH', 0.07253124, 'NXPI', 0.09813042, 'PYPL', 0.10145339, 'VIAB', 0.10335469, 'LBTYA', 0.114357226, 'LRCX', 0.11704111, 'NCLH', 0.14532024 |

Table 6: Variable importance ranking by DUAL and RETAIN methods on NASDAQ dataset.

| Dataset | Method | Rank of variables according to importance |
|---|---|---|
| NASDAQ | DUAL | 'NXPI', 0.003557, 'QCOM', 0.003564, 'FOX', 0.003566, 'NTAP', 0.003566, 'CELG', 0.003566, 'FOXA', 0.003567, 'PAYX', 0.003567, 'AAPL', 0.003567, 'WFM', 0.003567, 'ADSK', 0.003567, 'SBUX', 0.003567, 'STX', 0.003567, 'AKAM', 0.003567, 'DISH', 0.003567, 'AVGO', 0.003567, 'XLNX', 0.003567, 'AAL', 0.003567, 'FAST', 0.003567, 'TMUS', 0.003567, 'LRCX', 0.003567, 'NCLH', 0.003567, 'MCHP', 0.003567, 'MSFT', 0.003567, 'MU', 0.003567, 'NFLX', 0.003567, 'NVDA', 0.003567, 'PCAR', 0.003567, 'SIRI', 0.003567, 'MAR', 0.003567, 'TXN', 0.003567, 'ROST', 0.003567, 'CMCSA', 0.003567, 'ADI', 0.003567, 'ADP', 0.003567, 'DISCA', 0.003567, 'AMAT', 0.003567, 'WDC', 0.003567, 'CSX', 0.003567, 'WBA', 0.003567, 'GOOGL', 0.003622, 'COST', 0.003678, 'INTC', 0.003712, 'CTSH', 0.003908, 'BBBY', 0.004027, 'TRIP', 0.004881, 'MAT', 0.004956, 'ATVI', 0.005121, 'LBTYK', 0.00523, 'CERN', 0.00524, 'CTRP', 0.005283, 'ALXN', 0.00536, 'VOD', 0.005369, 'VRTX', 0.005433, 'LBTYA', 0.005445, 'MXIM', 0.00554, 'BIIB', 0.005554, 'EBAY', 0.005555, 'BIDU', 0.005605, 'FB', 0.005654, 'VIAB', 0.005685, 'GILD', 0.005695, 'AMGN', 0.005716, 'MYL', 0.005737, 'YHOO', 0.006166, 'KHC', 0.006555, 'AMZN', 0.006605, 'CSCO', 0.007836, 'ESRX', 0.010614, 'SWKS', 0.012777, 'MDLZ', 0.017898, 'CA', 0.02198, 'EXPE', 0.024373, 'QVCA', 0.026462, 'EA', 0.027808, 'TSLA', 0.043082, 'ADBE', 0.043829, 'JD', 0.071079, 'SYMC', 0.081596, 'PYPL', 0.087612, 'DLTR', 0.119737, 'TSCO', 0.122887 |
| | RETAIN | 'DLTR', 0.000866, 'QVCA', 0.001128, 'TSLA', 0.00119, 'PYPL', 0.00128, 'EA', 0.001439, 'EXPE', 0.001502, 'CA', 0.001713, 'TSCO', 0.001737, 'SYMC', 0.002334, 'ADBE', 0.00252, 'JD', 0.002607, 'AMZN', 0.003367, 'CSCO', 0.003543, 'KHC', 0.003996, 'CTSH', 0.004695, 'NXPI', 0.004865, 'EBAY', 0.004963, 'SWKS', 0.005011, 'MXIM', 0.005135, 'MYL', 0.005541, 'COST', 0.006052, 'BIDU', 0.006534, 'GOOGL', 0.006906, 'INTC', 0.007153, 'GILD', 0.007212, 'ESRX', 0.007512, 'NTAP', 0.007695, 'QCOM', 0.008037, 'CELG', 0.008168, 'MDLZ', 0.008829, 'AMGN', 0.008998, 'FOX', 0.009943, 'VIAB', 0.010123, 'AAPL', 0.010157, 'FB', 0.010359, 'YHOO', 0.010744, 'PAYX', 0.010899, 'BBBY', 0.01117, 'AKAM', 0.012054, 'BIIB', 0.012069, 'NFLX', 0.012266, 'ADSK', 0.012319, 'DISH', 0.012338, 'LBTYA', 0.012697, 'FOXA', 0.01282, 'MCHP', 0.012833, 'WFM', 0.012869, 'STX', 0.012887, 'VRTX', 0.013318, 'SBUX', 0.013458, 'VOD', 0.013798, 'ALXN', 0.013878, 'CTRP', 0.013963, 'SIRI', 0.01475, 'CERN', 0.014777, 'LBTYK', 0.014799, 'ATVI', 0.015651, 'AVGO', 0.016382, 'CMCSA', 0.016531, 'TXN', 0.016977, 'LRCX', 0.017131, 'AMAT', 0.017378, 'ROST', 0.017399, 'MU', 0.018045, 'TRIP', 0.018236, 'MAT', 0.018297, 'Auto-regressive', 0.018626, 'WDC', 0.019083, 'DISCA', 0.019233, 'FAST', 0.019392, 'CSX', 0.019734, 'WBA', 0.019984, 'AAL', 0.021188, 'ADI', 0.021215, 'NCLH', 0.022932, 'NVDA', 0.022994, 'TMUS', 0.024187, 'MSFT', 0.026354, 'ADP', 0.028515, 'MAR', 0.028783, 'PCAR', 0.029459, 'XLNX', 0.03248 |

Table 7: Variable importance ranking on PLANT and SML datasets.

| Dataset | Method | Rank of variables according to importance |
|---|---|---|
| PLANT | IMV-Full | 'Dew-point', 0.040899094, 'Wind-bearing', 0.04476319, 'Pressure', 0.06180005, 'P-temperature', 0.07244386, 'Auto-regressive', 0.1083069, 'Temperature', 0.11868146, 'Irradiance', 0.12043289, 'Humidity', 0.13192631, 'Cloud-cover', 0.14283147, 'Wind-speed', 0.15791483 |
| | IMV-Tensor | 'Dew-point', 0.034108493, 'Temperature', 0.041016363, 'Cloud-cover', 0.07639352, 'Irradiance', 0.08453229, 'Humidity', 0.09652364, 'Pressure', 0.10533282, 'Wind-speed', 0.115569875, 'P-temperature', 0.12627581, 'Auto-regressive', 0.15163974, 'Wind-bearing', 0.16860741 |
| | DUAL | 'Irradiance', 0.06128826, 'Dew-point', 0.066655099, 'Temperature', 0.071131147, 'Wind-speed', 0.094427079, 'Wind-bearing', 0.106529392, 'P-temperature', 0.115000054, 'Pressure', 0.115962856, 'Cloud cover', 0.144996881, 'Humidity', 0.224009201 |
| | RETAIN | 'Dewpoint', 0.031317, 'Temperature', 0.037989, 'Wind-bearing', 0.044226, 'Wind-speed', 0.052027, 'P-temperature', 0.053034, 'Cloud cover', 0.138427, 'Irradiance', 0.142899, 'Auto-regressive', 0.143269, 'Humidity', 0.172893, 'Pressure', 0.183919 |
| SML | IMV-Full | 'Outdoor temp.', 0.008530081, 'Outdoor humidity', 0.0120737655, 'Sun irradiance', 0.012943255, 'CO2 dining', 0.01563413, 'Sunlight in south', 0.01569774, 'Sun dusk', 0.015769556, 'Wind', 0.015868865], 'Forecast temp.', 0.015990425, 'Sunlight in west', 0.01609429, 'Lighting dining', 0.016338758, 'Humid. dining', 0.016379833, 'Sunlight in east', 0.016386982, 'Auto-regressive', 0.016530316, 'Temp. dining', 0.01663947, 'Lighting room', 0.18322693, 'CO2 room', 0.26715645, 'Humid. room', 0.33873916 |
| | IMV-Tensor | 'Temp. dining', 0.00178118, 'Auto-regressive', 0.0019085788, 'Lighting room', 0.0019707098, 'Outdoor humidity', 0.0019725773, 'CO2 room', 0.0019752455, 'Wind', 0.0019779552, 'Sunlight in south', 0.0019810565, 'Sunlight in east', 0.0019821296, 'Sun dusk', 0.001982501, 'Humid. room', 0.0020072663, 'Sunlight in west', 0.0028204536, 'Outdoor temp.', 0.0031943072, 'CO2 dining', 0.0056894314, 'Forecast temp.', 0.07802243, 'Humid. dining', 0.1383277, 'Lighting dining', 0.26305506, 'Sun irradiance', 0.48935142 |
| | DUAL | 'Humid. room', 0.059424, 'Humid. dining', 0.059656, 'Outdoor humidity', 0.059803, 'Temp. dining', 0.059878, 'Sun dusk', 0.060408, 'Sunlight in south', 0.061626, 'Wind', 0.061629, 'Sunlight in east', 0.062792, 'Lighting room', 0.063381, 'Forecast temp.', 0.063503, 'Sunlight in west', 0.063832, 'CO2 room', 0.064149, 'CO2 dining', 0.064383, 'Sun irradiance', 0.064703, 'Lighting dining', 0.0651, 'Outdoor temp.', 0.065733 |
| | RETAIN | 'Humid. dining', 0.012169, 'Humid. room', 0.014563, 'Sunlight in south', 0.018446, 'Lighting room', 0.018732, 'Outdoor humidity', 0.019388, 'Sunlight in west', 0.02219, 'Sunlight in east', 0.036744, 'CO2 room', 0.036864, 'CO2 dining', 0.037174, 'Sun dusk', 0.040011, 'Sun irradiance', 0.04075, 'Wind', 0.041191, 'Lighting dining', 0.054166, 'Forecast temp.', 0.133079, 'Outdoor temp.', 0.144314, 'Auto-regressive', 0.164673, 'Temp. dining', 0.165543 |

6.2.4    VARIABLE IMPORTANCE FOR PREDICTION

In this part, we add a new group of experiments using Pearson correlation, to compare with the variable importance learned by IMV-LSTM.

In particular, the Pearson correlation is used to pre-select the top $50\%$ variables with the highest (absolute) correlation values to the target variable. Then, we train IMV-Full and IMV-Tensor on the data consisting of these chosen variables. For the ease of comparison, in Table 8 we put together the prediction errors of IMV-LSTM family on variables chosen by Pearson and learned importance. IMV-LSTM in these two group of experiments shares the same network size.

In Table 8, we observe that by feeding variables chosen by Pearson correlation, IMV-Full and IMV-Tensor both degrade in prediction. Pearson measures linear correlation. Selecting variables based on it neglects the potential non-linear correlation in data and does not necessarily collect the proper variables for neural networks to reach desirable prediction performance.

In IMV-LSTM, importance values are derived from the variable-wise temporal attention and variable attention in the loss function. Such attentions are jointly learned to minimize the loss function (i.e. to optimize the prediction performance) and thus the variables chosen based on these learned importance values give rise to relatively retained prediction performance of IMV-LSTM.

Table 8: RMSE and MAE with std. errors under top $50\%$ variables

| Dataset | NASDAQ | PLANT | SML |
|---|---|---|---|
| | Learned importance | | |
| IMV-Full | $0.26 \pm 0.01, 0.23 \pm 0.02$ | $162.14 \pm 0.10, \mathbf{128.51 \pm 0.12}$ | $0.015 \pm 0.001, 0.011 \pm 0.002$ |
| IMV-Tensor | $\mathbf{0.12 \pm 0.007, 0.10 \pm 0.01}$ | $\mathbf{157.64 \pm 0.14}, 128.86 \pm 0.13$ | $\mathbf{0.007 \pm 0.0005, 0.006 \pm 0.0003}$ |
| | Pearson correlation | | |
| IMV-Full | $0.33 \pm 0.02, 0.30 \pm 0.03$ | $165.04 \pm 0.08, 129.09 \pm 0.09$ | $0.016 \pm 0.001, 0.013 \pm 0.0009$ |
| IMV-Tensor | $0.19 \pm 0.01, 0.15 \pm 0.02$ | $161.98 \pm 0.11, 131.17 \pm 0.12$ | $0.013 \pm 0.0008, 0.009 \pm 0.0005$ |

6.3    DISCUSSION

In this part, we summarize the insights from the experiments.

**Prediction performance**    For multi-variable data, capturing individual variable's behaviors and their interaction is the key for both prediction and interpretation. Conventional hidden states in standard LSTMs consume the data from all input variables at each step, while our IMV-LSTM family decomposes the hidden states by defining variable data flows for each hidden state element.

In the experiments, IMV-Full and IMV-Tensor outperform baselines using the traditional hidden states. Multi-variable data potentially carries different dynamics. Conventional hidden states mix the data of all input variables, thereby failing to explicitly capture individual dynamics. In the multi-variable setting, these opaque hidden states are a burden to both prediction and interpretation.

On the contrary, IMV-Tensor models individual variables and then uses mixture attention to capture the variable interaction by variable-wise hidden states. It achieves superior prediction performance and enables the interpretability on both temporal and variable levels.

**Effectiveness of importance values**    For LSTM networks on multi-variable data, importance values inherently learned by the network are more suitable for retaining useful variables for predicting.

By choosing the variables based on the learned importance value, IMV-LSTM family mostly retains the prediction performance and presents lower prediction errors on two datasets. The importance value in IMV-LSTM is derived during the training and therefore it is able to effectively identify the variables used by IMV-LSTM to minimize the loss function, i.e. maximize the prediction accuracy.

Pearson correlation variable selection leads to the quality loss in prediction performance, i.e. higher errors. Pearson correlation measures the linear correlation and pre-selecting variables based on it neglects the potential non-linear correlation in data indispensable for LSTMs to capture.

