# OpenReview forum: "Exploring the interpretability of LSTM neural networks over multi-variable data"
_ICLR.cc/2019/Conference_

### Official Review · AnonReviewer1 · 2018-11-02
**This paper explores the interpretability of LSTM with multivariable data while providing accurate forecasts in the context of time series. The paper is interesting and addresses a relevant topic. But it has several drawbacks that need to be addressed.**

**Rating:** 6
**Confidence:** 5

**Review:**

The contributions of this paper are in the field of LSTM, where the authors explore the interpretability of LSTM with multivariate data obtained from various and disparate applications. To this end, the authors endow their approach with tensorized hidden states and an update process in order to learn the hidden states. Furthermore, the authors develop a mixture attention mechanism and a summarization methods to quantify the temporal and variable importance in the data. They validate the forecasting  and interpretability performance of their approach with experiments.

The parer is interesting, well structured and and clearly written. Also, the addressed topic of interpretability is pertinent. However, I have several concerns.

1. In the related work the authors state that “In time series analysis, prediction with exogenous variables is formulated as an auto-regressive exogenous model ” . This is not always right - it is not imperative to add the auto-regressive terms, this is optional and depends on the way we want to formulate our time series forecasting approach and the known constraints.
2. In section 3 — Interpretable Multi-Variable LSTM, by stacking exogenous time series and target series, the authors implicitly formulate their algorithms in a way to consider auto-regression. And I have several concerns with this for time series forecasting. Because, the past is not always a predictor of the future even - particularly in time series context and in industrial settings. And in the occasions where the past allows to predict the future we do not necessarily need to use LSTM to forecast (the notion of persistency in forecasting is enough).  Therefore, the power of LSTM in forecasting would have been convincing if you omit the target series in your multi-variable input.
3. In Network Architecture section the authors develop tensorized hidden state and an update scheme. This idea is interesting, I think it would also be good to know what is the algorithmic complexity of this approach?
4.  In section 3.3 the authors state that "In the present paper, we choose the simple normalized summation function eq.(9). " Could the authors justify the reason behind this choice? I am not convinced of the reason behind this, especially the authors mention, right after,  that "It is flexible to choose alternative functions for f_{agg}"

5. In the experiment section, concerning the prediction performance the authors present a table showing their results, I believe it would have been more compelling to present the prediction results with graphs showing the normalized cumulative errors, as an example.

6. With regard to the interpretation of the results, the authors show the variable importance as a function of the epoch number, it would be equally important to correlate the same figure with the associated prediction results/normalized cumulative errors as a function of the epoch number - this will allow to assess the importance of the interpretability.

I think it would be important to further justify the pertinence of this work in terms of interpretability (the statement in the introduction "the interpretability of prediction models is essential for deployment and
knowledge extraction" seems to be limited) for example what does it bring knowing the variance importance  as a function of the epoch number. As an example, the Pearson correlation coefficient can help select relevant features to a model, and restrict the number of inputs to the relevant ones - can we draw inspiration from this and explain what the authors are proposing in terms of interpretability... Here the idea is to have a motivation presenting the merits of this work, which I think is missing - particularly with the experiments presented here.

---

> ### Comment · AnonReviewer1 · 2018-12-10
> **I revised my rating on the basis of the improvements brought to the paper.**
>
> In light of the improvements brought to the paper to address some of the concerns initially raised, I believe the paper will be of interest to the ICLR community.

---

> > ### Author Response · Authors · 2018-12-12
> > **Thanks for the reply!**
> >
> > Dear Reviewer,
> >
> > Thanks for updating the rating!
> >
> > We are continuously working on improving the manuscript both theoretically and experimentally.
> >
> > Feel free to post comments if you have additional advice.
> >
> > Thanks!
> >
> > Best regards,
> > Authors

---

### Official Review · AnonReviewer3 · 2018-11-02
**Interesting and potent interpretable LSTM without an actual interpretation in terms of the problem: the model claims to make variable/temporal variable importance but without actually interpreting the quality.**

**Rating:** 5
**Confidence:** 3

**Review:**

This paper describes a recurrent model (LSTM specifically, but generalizable) which can produce variable-wise hidden states that can be further used for two types of attentions: 1) variable importance for the importance of each variable (not accounting for time), and 2) temporal importance of each variable for the importance of each variable over time. The proposed NN model (IMV-LSTM) does not seem to directly provide such importance. Rather, the outputs are “decomposed” for each variable/time that allows probabilistic inference on top of this.

One of my main concerns (described in Cons/Comments below) is how it is not straightforward to grasp the quality of variable importance and temporal variable importance results despite this is the key strength of this paper. If this comes from my lack of understanding, I would appreciate if the authors could provide a little more explanation.

Pros:
1.	The overall quality of the paper is decent and mostly clear.
2.	The experiments are quite extensive.
3.	The fact that each variable should have different level of importance is interesting and practical.

Cons/Comments:
1.	The term “tensor” is used throughout the paper to describe the stacked matrices. While this is not technically wrong to describe 2>-dimensional structures, this term could potentially imply (and make the readers to expect) tensor-based schemes such as tensor decomposition. This is not necessarily bad, but to me, “tensor” and “variable-wise correspondence” do not seems to be associated too deeply since the “tensor” used in IMV-LSTM is a stack of matrices that are also independently used with respect to each other.

2.	The variable importance experiments seem quite extensive and thorough, especially the lists of variable-wise temporal importance matrices provided in the appendix. However, the authors could provide some significance or relevance of the findings with respect to any domain knowledge or literature, it may help further appreciate and interpret the quality of the variable importance which is quite subjective to non-experts. Such information may not even need to be in the main paper; including a short description in the appendix.

3.	Related to comments (2), the difference between IMV-Full and IMV-Tensor is hard to interpret since neither one is always better than the other (i.e., IMV-Full > IMV-Tensor in some experiments, vice versa). While the key difference is speculated to be from how the LSTM handles the variables, I am curious how this related to the differences in the results and how the differences variable importance results (i.e., Fig.3) can be in at least speculated.

Questions:
1.	Should \tilde{h}_t in Figure 1 (a) be \tilde{h}__{t-1} since this hidden state is from t-1? The figure itself currently implies that the hidden state for t is used, but this is computed from x_t using U_j. With \tilde{h}__{t-1}, it follows Eq.(1).

2.	In Equation set 2 for IMV-Tensor, are W and U (not W_j and U_j) also in tensor forms so each variable and hidden state get transformed correspondingly (i.e., W_1 for h^1_{t-1}, U_1 for x^1_t).

3.	The IMV-Tensor version of IMV-LSTM (related to the question above) can be considered as a set of parallel LSTMs, one for each variable. Such independence could also be inferred from Figure 1. If that’s the case, where do the variables “interact” with each other? Is this happening in the later stage where the hidden states across variable/time are aggregated in the attention stage (Eq.(8) and on)?

4.	Up until Eq.(8), n was used for the variable index where n = 1,…,N. In Eq.(8), it seems to be still used as the variable index (i.e., h_T^n and g^n), but it is also a set of possible values for a random variable z_{T+1}. Is n used the same way for z_{T+1} as well? I am slightly confused on how z is used. Also, (just to clarify), if we use N variables, we are using y_t as well (i.e., [x_t^1,…,x_t^{N-1}, y_t])?

5.	f_agg: Is this for aggregating over instances? For \bar{\alpha}^n, I’m guessing this is aggregated over instances for variable n for t=1,…,T_1.

6.	I am not too familiar with the notion of “time-lag” in the experiments. If the authors could explain this a little bit, I would appreciate it.

---

### Official Review · AnonReviewer2 · 2018-11-03
**Nice work, but claims are bit much**

**Rating:** 6
**Confidence:** 5

**Review:**

Summary:
The authors propose IMV-LSTM, which can handle multi-variate time series data in a manner that enables accurate forecasting, and interpretation (importance of variables across time, and importance of each variable). The authors use one LSTM per variable, and propose two implementations: IMV-Full explicitly tries to capture the interaction between the variables before mixing the LSTM hidden layers with attention. IMV-Tensor uses separate LSTMs for each variable that remain separate, and mixes the hidden layers of the LSTMs using attention. The propose model outperforms popular interpretable models on three different datasets, and the experiments regarding the variable importance is convincing.

Pros:
- The paper is clearly written, easy to understand.
- IMV-LSTM outperforms many baselines including popular interpretable models on three different datasets, and the interpretation part is not super rigorous, but convincing enough.
- Multi-variate time-series data are very common, therefore an interpretable, accurate models such as IMV-LSTM have a big practical impact.
- I like the idea of using the important variables to train another model for testing how accurately the models can choose important variables

Issues:
- In the introduction: claim that attention mechanism can unveil the effect of variable to the target is tricky, potentially dangerous: Attention is attention. It is no causal, let alone correlation. Coefficients in logistic regression are correlated with the prediction target. Variables with high attention has "some relationship" with the prediction target.
- The methodological novelty of IMV-LSTM is limited. Using attention mechanism on RNN to provide interpretation has been explored quite often. This paper is not so different from other works [1,2,3]
- Claim that this is the first work to derive temporal-level & variable-level importance is not convincing: The importance calculation of this paper boils down to averaging the attention values. This can be easily done in the previous works [1,2,3], or any model that uses attention on each input channel and on the temporal axis.
- Can't follow Eq.10. How is this justified?


[1] Choi, E., Bahadori, M.T., Sun, J., Kulas, J., Schuetz, A. and Stewart, W., 2016. Retain: An interpretable predictive model for healthcare using reverse time attention mechanism. In Advances in Neural Information Processing Systems (pp. 3504-3512).
[2] Zhang, J., Kowsari, K., Harrison, J.H., Lobo, J.M. and Barnes, L.E., 2018. Patient2Vec: A Personalized Interpretable Deep Representation of the Longitudinal Electronic Health Record. IEEE Access.
[3] Xu, Y., Biswal, S., Deshpande, S.R., Maher, K.O. and Sun, J., 2018, July. RAIM: Recurrent Attentive and Intensive Model of Multimodal Patient Monitoring Data. In Proceedings of the 24th ACM SIGKDD International Conference on Knowledge Discovery & Data Mining (pp. 2565-2573). ACM.

---

> ### Comment · AnonReviewer2 · 2018-12-10
> **Retaining the rating**
>
> After reading the authors' feedback, I must say that I am still not convinced of the strong novelty of this work.
> The proposed method for deriving the importance (variable-wise or time-wise) is still, in essence, averaging the attention values.
> And the authors' feedback suggests that the authors may not have a clear understanding of RETAIN. What separates RETAIN from other attention-based models is that RETAIN provides a way to precisely calculate the contribution of each variable in each timestep, which is not the same as calculating the variable importance by the attention each variable receives.
> Therefore, when the authors were conducting the experiment in 4.5 using RETAIN, I wonder if the authors selected the variables by correctly calculating the contribution of each variable, or simply used the attention that each variable received.
> With these said, I still think the paper proposes a decent approach, and the overall quality of the paper calls for a 6, and I retain my rating.
> However, if this paper is accepted, I suggest that the authors clarify the points I raised regarding RETAIN, as imprecise description of baselines could lower the credibility of the entire paper (even though the paper's idea itself is nice).

---

> > ### Author Response · Authors · 2018-12-11
> > **About RETAIN**
> >
> > Dear Reviewer,
> >
> > Thanks for your reply to the revision! Maybe we did not explain clearly in the previous response.
> >
> > We understand and fully agree that RETAIN is innovative in calculating the contribution of each variable in each timestep.
> >
> > -- regarding RETAIN
> >
> > What we tried to explain is that the derivation of the attention at each time step (i.e. “Step 3” in the paper of RETAIN) and using the attention value to weight input (i.e. “Step 4”) are problematic.
> >
> > We would like to draw the attention of the community and inspire some insights into designing attention or contribution measures on multi-variable data.
> >
> > In particular, in the RETAIN paper, in “Step 3”, $\beta_j$ over variables is derived from the hidden states of RNN_{beta}. RNN_{beta} consumes multi-variable data in a conventional way and thus the hidden states mix information from all variables. Each element of $\beta_j$ is then derived from hidden states with mixed information and opaque data flows.
> >
> > Our hypothesis is that it is improper to use each element of $\beta_j$ to represent the contribution or importance measure of corresponding variables at each timestep, since each element includes the mixed contribution of all input variables.
> >
> > As for “Step 4”, using attention value to directly weight the input data could be problematic as well if we take into account the correlation direction and domain of the input variables.
> >
> > -- “Therefore, when the authors were conducting the experiment in 4.5 using RETAIN, I wonder if the authors selected the variables by correctly calculating the contribution of each variable, or simply used the attention that each variable received.”
> >
> > The experiments in 4.5 are strictly in accordance with RETAIN paper and use the “contribution coefficient” defined in Eq. (5) in RETAIN paper to select variables.

---

### Meta-Review · Area_Chair1 · 2018-12-14
**Borderline, with no clear reviewer endorsement**

**Confidence:** 3
**Recommendation:** Reject

**Metareview:**

The reviewers appreciated the clarity of writing, and the importance of the problem being addressed. There was a moderate amount of discussion around the paper, but the two reviewers who responded to the author discussion were split in their opinion, with one slightly increasing their score to a 6, and the other remaining unconvinced. The scores overall are borderline for ICLR acceptance, and given that, no reviewer stepped forward to champion the paper.